# Association between Preoperative Retrograde Hepatic Vein Flow and Acute Kidney Injury after Cardiac Surgery [note 1]

**DOI:** 10.3390/diagnostics12030699

**Published:** 2022-03-12

**Authors:** Csaba Eke, András Szabó, Ádám Nagy, Boglár Párkányi, Miklós D. Kertai, Levente Fazekas, Attila Kovács, Bálint Lakatos, István Hartyánszky, János Gál, Béla Merkely, Andrea Székely

**Affiliations:** 1Károly Rácz School of PhD Studies, Semmelweis University, 1085 Budapest, Hungary; 25csabaeke@gmail.com (C.E.); andraas.szaboo@gmail.com (A.S.); nagyadam05@gmail.com (Á.N.); 2Faculty of Medicine, Semmelweis University, 1085 Budapest, Hungary; parkanyib@gmail.com; 3Department of Anesthesiology, Vanderbilt University Medical Center, Nashville, TN 37212, USA; miklos.kertai@vumc.org; 4Heart and Vascular Center, Semmelweis University, 1085 Budapest, Hungary; drfalev@gmail.com (L.F.); kovacs.attila@med.semmelweis-univ.hu (A.K.); lakatos.balint@med.semmelweis-univ.hu (B.L.); hartyanszky.istvan@gmail.com (I.H.); merkely.bela@med.semmelweis-univ.hu (B.M.); 5Department of Anesthesiology and Intensive Therapy, Semmelweis University, 1085 Budapest, Hungary; gal.janos@med.semmelweis-univ.hu

**Keywords:** doppler ultrasound, heart failure, acute kidney injury, hepatic venous flow

## Abstract

Key questions: Is there a predictive value of hepatic venous flow patterns for postoperative acute kidney injury (AKI) after cardiac surgery? Key findings: In patients who underwent cardiac surgery, retrograde hepatic venous waves (A, V) and their respective ratio to anterograde waves showed a strong association with postoperative AKI, defined as the percentage change of the highest postoperative serum creatinine from the baseline preoperative concentration (%ΔCr). The velocity time integral (VTI) of the retrograde A wave and the ratio of the retrograde and anterograde waves’ VTI were independently associated with AKI after adjustment for disease severity. Take-home message: A higher ratio of retrograde/antegrade waves in hepatic venous retrograde waves, which are related to hepatic stasis, may predict AKI after cardiac surgery. Introduction: Hepatic venous flow patterns reflect pressure changes in the right ventricle and are also markers of systemic venous congestion. Pulsatility of the inferior caval vein was used to predict the risk of acute kidney injury (AKI) after cardiac surgery. Aims: Our objective was to evaluate the association between preoperative hepatic venous flow patterns and the risk of AKI in patients after cardiac surgery. Methods: This prospective, observational study included 98 patients without preexisting liver disease who underwent cardiac surgery between 1 January 2018, and 31 March 2020, at a tertiary heart center. In addition to a routine echocardiographic examination, we recorded the maximal velocity and velocity time integral (VTI) of the standard four waves in the common hepatic vein with Doppler ultrasound. Our primary outcome measure was postoperative AKI, defined as the percentage change of the highest postoperative serum creatinine from the baseline preoperative concentration (%ΔCr). The secondary outcome was AKI, defined by KDIGO (Kidney Disease Improving Global Outcomes) criteria. Results: The median age of the patients was 69.8 years (interquartile range [IQR 25–75] 13 years). Seventeen patients (17.3%) developed postoperative AKI based on the KDIGO. The VTI of the retrograde A waves in the hepatic veins showed a strong correlation (B: 0.714; *p* = 0.0001) with an increase in creatinine levels after cardiac surgery. The velocity time integral (VTI) of the A wave (B = 0.038, 95% CI = 0.025–0.051, *p* < 0.001) and the ratio of VTI of the retrograde and anterograde waves (B = 0.233, 95% CI = 0.112–0.356, *p* < 0.001) were independently associated with an increase in creatinine levels. Conclusions: The severity of hepatic venous regurgitation can be a sign of venous congestion and seems to be related to the development of AKI.

## 1. Introduction

Perioperative acute kidney injury (AKI) is common and is associated with considerable morbidity and mortality after cardiac surgery. Several risk factors have been identified in the development of AKI, such as low output syndrome, cardiopulmonary bypass, and chronic kidney disease [1]. AKI [2] has been associated with an increased risk of sepsis, anemia, coagulopathy, and prolonged mechanical ventilation [3].

More recently, hepatic venous congestion and the consequent hepatic parenchymal dysfunction have been found to be associated with increased risk for mortality and morbidity in patients with end-stage cardiac failure [2,4,5]. In cardiac surgery patients with congestive heart failure, venous hypertension may worsen kidney function by reducing the effective transrenal gradient, decreasing the glomerular filtration rate, and leading to fluid overload in the postoperative period. In a pathophysiological aspect, liver damage is caused by stasis or low cardiac output [6]. Low cardiac output causes ischemic hepatitis with consequent centrilobular necrosis, leading to elevated transaminase and bilirubin levels in the blood. Chronic congestion is characterized by high alkaline phosphatase (ALP) and GGT levels. There have been several studies that examined the potential role of cardiogenic liver failure and kidney injury their crucial role in long-term survival [7,8,9], however, these studies failed to examine the degree and extent of hepatic venous congestion and the consequent hepatic parenchymal dysfunction on the risk for postoperative AKI after cardiac surgery.

We hypothesized that the analysis of the hepatic vein flow profile and calculation of the anterograde and retrograde flow before cardiac surgery could be helpful markers in the prediction of postoperative AKI. Additionally, we compared the ratio of retrograde flow with preoperative renal and hepatic laboratory parameters and with postoperative complications.

## 2. Methods

### 2.1. Study Design

The study results are reported according to the STROBE statement. Our study has approval from the Institutional Review Board of Semmelweis University (IRB 141/2018), and it is registered as ClinicalTrials.gov number NCT02893657. In this prospective, observational study, we enrolled 98 patients who underwent cardiac surgery between January 2018 and December 2019 in a tertiary heart center. Patients who had preoperative chronic kidney disease (defined as GFR under 30 mL/min/1.73 m^2^ hepatic cirrhosis, or portal vein thrombosis were excluded. Each patient who agreed with the participation signed the informed consent before the first investigation, which was usually two to three days before the planned surgery.

### 2.2. Definitions and Measurements (Variables and Data Sources and Grouping)

Demographic data, preoperative laboratory parameters, intraoperative variables (procedure, cardiopulmonary bypass time, fluid balance, administration of blood products, need for vasoactive medication), and postoperative factors (fluid balance, vasoactive medications) were used in the analysis. In particular, we collected information on the predictors of the European System for Cardiac Operative Risk Evaluation [10,11] and the MELD Model for End-stage Liver Disease (MELD) score [12,13], vasoactive-inotrope score (VIS), and inotropic scores (IS) [14]. We have calculated the AKI score Thakar et al. Subsequently these scores were estimated and adjusted for in our multivariable analyses. The list of the collected perioperative and demographic variables and the number of the missing values are shown in the Appendix A.

### 2.3. Analyses of the Hepatic Veins

Venous blood flow in the common hepatic vein immediately before influx into the inferior caval vein was used for the analysis. A significant right inferior hepatic vein complements the right hepatic vein in 30–61% of cases. The left and middle hepatic veins join to form a single vein before entering the inferior vena cava (IVC) in 60–86% of people [15,16]. The normal hepatic vein waveform, despite commonly being described as triphasic, has four components: a retrograde A wave, an antegrade S wave, a transitional V wave (which may be antegrade, retrograde, or neutral), and an antegrade D wave [17]. We recorded the standard four waves’ (A, S, V, D) maximal velocity and velocity-time integral (VTI) [18,19]. The ratio of maximum to retrograde compared to anterograde velocity and the ratio of retrograde VTI compared to anterograde VTIs were calculated [20,21]. The hepatic vein waves and the analyzed waves are shown in Figure 1, Figure 2, Figure 3 and Figure 4.

The echocardiographic investigation was performed by two cardiologists. Standard 2D parameters were ejection fraction, tricuspidal anular plane systolic excursion, the diameters of atria and ventricles, and the occurrence of valve insufficiency. The ultrasound examinations were performed by board-certified cardiologists in echocardiography and were recorded on the same machine and analyzed by the same person after completion of the study. The physicians in the postoperative period were blinded to the results of the hepatic flow measurements.

### 2.4. Outcomes

Our primary outcome was postoperative AKI, defined as the percentage change of the highest postoperative serum creatinine from the baseline preoperative concentration (%ΔCr) on the first three postoperative days. Baseline creatinine was defined as the preoperative creatinine level measured after the indexed hospital admission but before the surgery. The peak fractional change was used as a marker of renal filtration impairment [22,23]. The secondary outcome was AKI, defined by KDIGO (Kidney Disease Improving Global Outcomes) criteria [24,25].

### 2.5. Power Calculation

Our power analysis indicated that with 8 potential predictors of acute kidney injury, a minimum of 90 patients were needed to perform a prediction model with adequate sample size and avoiding overfitting.

### 2.6. Statistical Analysis

Normality was tested with the Kolmogorov–Smirnov test. Skewed distributions are described as medians and interquartile ranges (interquartile range 25–75) and were compared using the Mann–Whitney U test. Descriptive statistics for serum creatinine concentrations, estimated GFR based on the equation, as well as KDIGO based AKI definitions are provided for both baseline (preoperative) and peak postoperative measures. Continuous variables were first expanded with restricted cubic splines (to allow for potentially nonlinear effects) and were only used in linear form if the deviation from linearity was not significant as indicated by the global F test (*p* > 0.05). Our analyses indicated that AVmax, A VTI, and antero/reroVTI were non-linear variables, and therefore, these variables were transformed before entering them to the multivariable regression analyses. The effect of hepatic waves on the peak creatinine fractional change (%ΔCr) was evaluated using multivariable linear regression analysis and adjusted for patient- and surgery-related characteristics that included adjustment for significant perioperative variables [26]. The following variables were considered, including age, renal function, EuroSCORE II and AKI prediction score by Thakar et al. [23], operation time, fluid balance, and maximum vasopressor-inotropics score in the first 72 postoperative hours. The variables selection was based on a set of previously published studies with a focus on AKI after cardiac surgery. The multivariable model was tested for multicollinearity (by volume inflation factors) between these clinical variables. The adjusted changes in R^2^ were reported after each step of the regression model to determine the contribution of each additional variable that was added to the regression model.

Statistical tests were 2-sided, and *p* < 0.05 was considered statistically significant. Statistical analyses were performed with SPSS software, Version 27.0 (IBM, Armonk, NY, USA).

## 3. Results

### Descriptive Data

Of the 98 patients, 66 (67%) were males. The median age of the patients was 69.8 years (interquartile range [IQR 25–75] 13 years). The baseline and clinical characteristics of the study population are shown in Table 1. The most common type of cardiac surgery was coronary artery bypass grafting (CABG) surgery followed by aortic valve repair. The average change in postoperative creatinine levels compared to baseline was an 8.7% increase from 83.2 µmol/L (IQR 25–75: 7.8) to 91.9 µmol (IQR 25–75: 12.8). In patients with AKI (*n* = 17), there was a 72% increase from 78.4 µmol/L (IQR 25–75: 8.7) to 137.0 µmol/L (IQR 25–75: 32.7), and the ratio of postoperative to preoperative creatinine was 1.65 (1.51–1.81). Seventeen patients (17.3%) developed AKI according to the KDIGO criteria.

There was no difference between the AKI and non-AKI patients in standard echocardiographic parameters before and after surgery. Using linear regression analysis, the right atrium systolic area showed a significant correlation with ΔCr (B: −0.004 *p* = 0.045), as seen in Appendix A. The VTI of the A wave was associated with preoperative bilirubin levels (B: 0.51 *p* = 0.003).

With univariable linear regression, we calculated the correlation between the preoperative parameters and the changes in postoperative creatinine levels compared to baseline (Table 2 and Appendix A). In the multivariable regression model, VTI and Vmax of the A wave correlated independently with the increase in the creatinine level. The VTI of the retrograde waves (A + D) and the VTI of the retrograde/anterograde waves ratio were independently associated with the increase in creatinine levels (Table 2).

Maximum velocity (vmax) and velocity-time-integral (VTI) of A, S, V, and D waves were analyzed with creatinine change in univariable and multivariable regression models. We also calculated the sum of retrograde (retro VTI), the sum of anterograde flow (anteroVTI), and the ratio of retrograde and anterograde waves.

The variables were adjusted for age, EuroSCORE (European System for Cardiac Operative Risk Evaluation), AKI (acute kidney injury by Thakar et al.) score, operation time, fluid balance, and vasoactive inotropic score (VIS).

R^2^ adj. = 0.157 for inclusion of age, EuroSCORE, AKI score, operation time, fluid balance, and VIS, R^2^ adj. = 0.472 for inclusion of A VTI, F change *p* < 0.0001.

R^2^ adj. = 0.155 for inclusion of age, EuroSCORE, AKI score, operation time, fluid balance, and VIS, R^2^ adj. = 0.401 for inclusion of A vmax, F change *p* < 0.0001.

R^2^ adj. = 0.157 for inclusion of age, EuroSCORE, AKI score, operation time, fluid balance, and VIS, R^2^ adj. = 0.241 for inclusion of retrograde VTI, F change *p* < 0.0001.

R^2^ adj. = 0.157 for inclusion of age, EuroSCORE, AKI score, operation time, fluid balance, and VIS, R^2^ adj. = 0.437 for inclusion of ratio VTI, F change *p* < 0.0001.

In the ROC analysis, the VTI ratio of anterograde to retrograde waves had a 0.654 discriminative value for the prediction of AKI. We observed a relationship between the severity of KDIGO stages and the VTI ratio (*p* = 0.045). Among the postoperative variables characterizing the complications, length of mechanical ventilation and need for furosemide in the first 72 h were associated with changes in postoperative creatinine levels (ΔCr), as seen in Appendix A.

## 4. Discussion

In patients undergoing cardiac surgery, we found a significant correlation between the ratio of retrograde waves compared to anterograde hepatic venous waves and postoperative AKI. The strong relationship of the retrograde A wave and the retrograde to anterograde ratio with AKI remained statistically significant after adjusting for patient- and procedure-related characteristics. VTI, but not maximum velocity of the retrograde and anterograde waves, showed a significant correlation with AKI.

Echocardiography is a widely used and accepted technique in the perioperative management of patients undergoing cardiac surgery [27,28]. Recently, several point-of-care ultrasound markers (POCUS) [29] have been identified as potential markers to detect venous congestion and fluid overload [30]. The flow patterns of caval, hepatic, and renal veins have been studied in this regard, and pulsatility or maximum velocities of the retrograde and anterograde waves have been found to be associated with the severity of congestive heart failure after heart surgery [31,32]. The clinical usefulness of POCUS in detecting significant venous congestion may be strengthened by quantification of the measured waves in the renal and hepatic veins and by the comparison of hepatic and renal laboratory parameters of the actual measurements [30]. We have supposed that the baseline hepatic vein pattern can be used to predict abdominal venous insufficiency, which can lead to decreased transrenal perfusion gradients and abdominal compartment syndrome. Moreover, VTI of the retrograde wave was associated with higher bilirubin levels, indicating disturbed hepatic excretory function.

The severity of hepatic venous congestion was found to be a predictor of AKI in a prospective study among cardiac surgery patients [30,33,34]. Various studies have measured the maximum velocity of the S and D waves and the severity of venous congestion by the S and D wave ratios preoperatively, during intensive care, and after discharge [32,35]. They also found a significant association between AKI and the preoperative maximum velocity of S waves. In contrast, we measured both maximum velocity and VTI of the hepatic waves and found that VTI, and their derivates, were associated with the postoperative elevation of creatinine and AKI while maximum velocity was not. Based on our results, we found that the magnitude of the retrograde waves in hepatic venous circulation is not solely attributed to the severity of right heart failure. Among these parameters (tricuspid regurgitation, TAPSE, or right ventricular diameters) only the right atrial systolic area showed an association with the retrograde waves of the hepatic vein.

The link between cardiovascular diseases and the liver has been a highlighted [36,37]. The progression of cardiac failure causes hypoperfusion and stasis in the liver, and it leads to severe dysfunction [15,38]. Calculation of MELD and their modifications [12] has been associated with increased mortality and morbidity in end-stage heat failure and in patients after heart transplantation [2]. In our study population, there were no differences between MELD scores in AKI and non-AKI patients, but the association between bilirubin and retrograde A waves draws attention to possible hepatic dysfunction in cases of fluid overload and vasoactive drug administration [39]. In patients with univentricular physiology (Fontane circulation), early weaning from mechanical ventilation, and return of spontaneous breathing is crucial, since negative thoracic pressure can increase venous return, increase the preload, and maintain cardiac output [40]. The scenario is similar: negative pressures (i.e., early extubation) can help to decrease the afterload of the liver, while fluid overload, prolonged ventilatory support, and edema formation will worsen venous congestion. Early recovery after heart surgery can be promoted by optimal fluid management, and the preoperative detection of venous congestion or venous abdominal insufficiency with adequate therapy can be successfully treated in the majority of cases.

Acute kidney injury is associated with high morbidity and mortality after cardiac surgery. We chose AKI and an increase in creatinine levels as the main outcomes in our study [1,30,41]. In the multivariable model, we also adjusted the retrograde waveforms for EuroSCORE II, GFR, and operative factors, and it remained significant, but the occurrence of AKI was relatively low in our study population. Therefore, we have focused on postoperative creatinine elevation (even in cases that do not reach KDIGO criteria), which has yielded relevant results in the association between creatinine increase and discrete signs of venous congestion. Our results indicate that the severity of venous congestion should be evaluated preoperatively, but it is also advised to follow up in the postoperative period.

Our study has limitations. First, it was a single center study. In the planning of the study, we did not think about continuing it in the postoperative period. A couple of perioperative factors, such as positive fluid balance, vasopressor use, or positive settings on mechanical ventilation, can significantly influence the anterograde/retrograde ratios. Therefore, preoperative baseline images of the hepatic veins must be obtained in spontaneously breathing patients. Most likely, the anterograde/retrograde ratio, which excludes the bias caused by the different axes of the Doppler wave on the hepatic vein, would be the optimal parameter.

## 5. Conclusions

We can conclude that congestion and/or abdominal venous insufficiency in the liver can predict worse renal function. Abdominal ultrasound is an easily available and, most importantly, noninvasive tool that should be used to detect the signs of abdominal venous congestion. Monitoring the hepatic veins can easily be performed, and in conjunction with standard preoperative echocardiography, it may yield additional information about the liver and kidney than solely laboratory parameters. It is also advised to quantitatively express our measures and not only with flow pattern types.

## Figures and Tables

**Figure 1 diagnostics-12-00699-f001:**
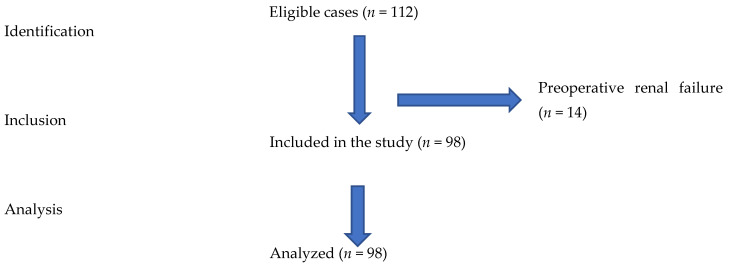
Flow chart.

**Figure 2 diagnostics-12-00699-f002:**
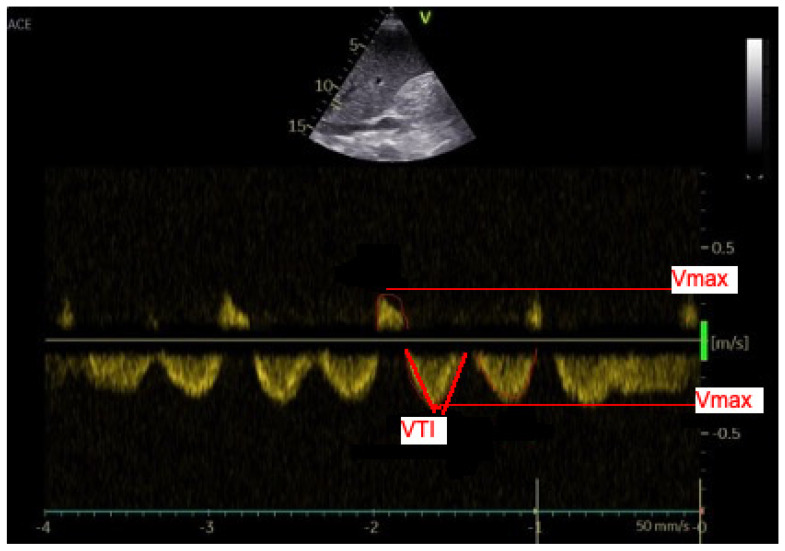
Doppler US analysis of the hepatic veins–maximum velocity and velocity-time integral were measured (a line was adjusted to the maximum point and continuous wave Doppler’s spectral curve was calculated by the machine. A and V waves are retrograde, S and D are anterograde waves).

**Figure 3 diagnostics-12-00699-f003:**
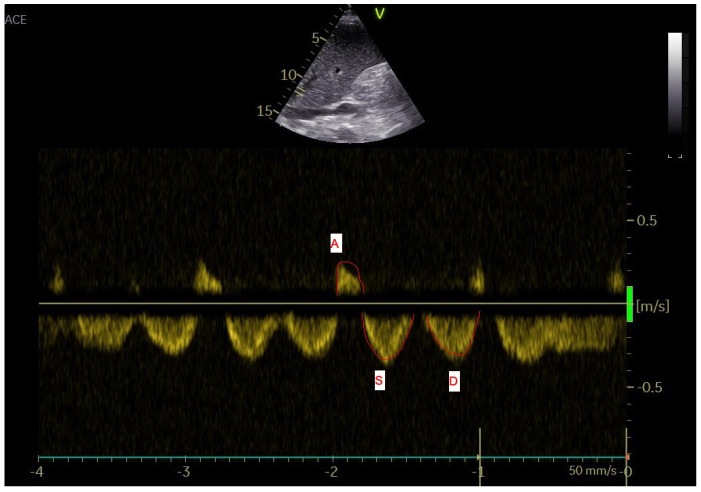
Doppler ultrasound analysis of the hepatic veins–maximum velocity and velocity-time integral were measured (a line was adjusted to the maximum point and continuous wave Doppler’s spectral curve was calculated by the machine. A and V waves are retrograde, S and D are anterograde waves).

**Figure 4 diagnostics-12-00699-f004:**
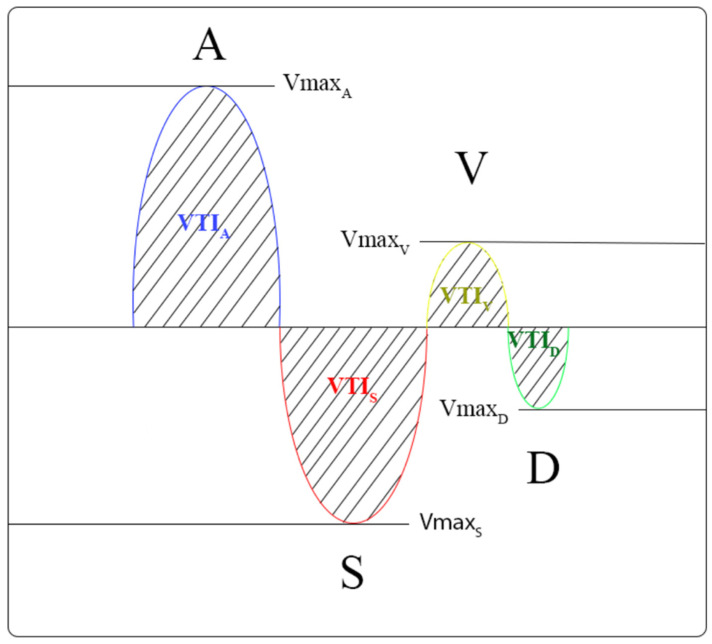
Schematic presentation of maximum velocity and velocity-time-integral of the hepatic waves (maximum velocity’s point and VTI’s area, A and V waves are retrograde, S and D are anterograde waves).

**Table 1 diagnostics-12-00699-t001:** Demographic and clinical data of the population and the AKI and non-AKI subgroup.

	All Patients	AKI	Non-AKI	*p*
Male	66 (67%)	11 (62%)	55 (70%)	0.43
Female (Nr)	32 (33%)	6 (38%)	26 (30%)	0.16
Age (years)	68.8 (11.2)	69.1 (7.4)	63.5 (13.9)	0.015
Diabetes	20 (20%)	5 (25%)	15 (19%)	0.09
NYHA III/IV	41 (41%)	10 (58.8%)	31 (39%)	0.12
EuroSCORE	1.6 (0.9)	1.6 (1.0)	1.5 (0.7)	0.09
Weight (kg)	74.6 (8.1)	72.6 (7.1)	75.1 (9.12)	0.43
Operation time (min)	182.4 (39.1)	178.1 (41.1)	188. 8 (39.1)	0.88
Aorta cross clamp time (min)	47.8 (7.1)	40.8 (9.1)	48.1 (7.6)	0.73
Type of surgery				
AVR	28 (29%)	6 (35%)	22 (26%)	0.11
CABG	39 (40%)	7 (41%)	32 (39%)	0.06
MVR	20 (20%)	3 (17%)	17 (20%)	0.09
Combined	12 (12%)	1 (7%)	11 (15%)	0.12
Hemoglobin (g/L)	137.1 (18.5)	135.3 (19.7)	138.4 (18.7)	0.61
Albumin (g/L)	41.4 (8.6)	38.8 (8.1)	43.7 (7.9)	0.83
Platelets (1000/L)	217.5 (62.2)	218.6 (64.9)	211.4 (58.7)	0.75
CRP (mg/L)	4.3 (3.3)	4.2 (3.9)	4.5 (3.3)	0.53
INR	1.9 (10.5)	1.1 (0.3)	3.9 (22.2)	0.34
ASAT (U/L)	20.9 (10.4)	18.8 (6.3)	21.2 (13.2)	0.34
ALAT (U/L)	31.4 (21.3)	34.2 (18.9)	30.4 (27.3)	0.56
Creatinine (µmol/L)	87.8 (20.1)	109.8 (25.1)	78.4 (17.7)	0.001
Urea Nitrogen (mmol/L)	6.4 (2.1)	5.9 (1.8)	7.4 (3.2)	0.18
GFR (ml/min/1.73 )	79.7 (15.4)	80.2 (15.6)	76.0 (16.6)	0.004
Bilirubin (µmol/L)	17.8 (43.3)	11.4 (7.4)	21.8 (76.5)	0.31
MELD	7.1 (19.0)	7.50 (4.6)	5.3 (1.2)	0.50
NYHA	2.1 (0.5)	2.1 (0.4)	2.2 (0.5)	0.10
EuroSCORE	1.6 (0.9)	1.6 (0.9)	1.5 (1.0)	0.09

AKI: Acute kidney injury, AVR: Aortic valve repair, MVR: Mitral valve repair, CABG: Coronary artery bypass grafting, CRP: C-reactive protein, EuroSCORE: European System for Cardiac Operative Risk Evaluation INR: Internationally normalized ratio, ASAT: Aspartate aminotransferase, ALAT: Alanine aminotransferase, GFR: Glomerular filtration rate, MELD: Model for end-stage liver disease.

**Table 2 diagnostics-12-00699-t002:** Results of the regression models for postoperative creatinine change.

	Univariable Linear Regression				Multivariable Regression			
	B	95% CI		*p* Value	B	95% CI		*p* Value
Avmax (m/s)	0.640	0.332	0.948	<0.001	0.714	0.437	0.991	<0.001
A VTI (cm)	0.035	0.021	0.050	<0.001	0.038	0.025	0.051	<0.001
Svmax (m/s)	0.049	−0.153	0.251	0.631				
S VTI (cm)	−0.002	−0.011	0.007	0.691				
Vvmax (m/s)	0.075	−0.127	0.277	0.462				
V VTI (cm)	0.011	−0.003	0.025	0.127				
Dvmax (m/s)	0.151	−0.163	0.466	0.342				
D VTI (cm)	0.010	−0.005	0.024	0.177				
Anterovmax (m/s)	0.062	−0.080	0.204	0.388				
Retrovmax (m/s)	0.168	−0.009	0.345	0.062				
Ratiovmax	0.091	−0.124	0.307	0.402				
AnteroVTI (cm)	0.001	−0.006	0.009	0.690				
RetroVTI (cm)	0.017	0.006	0.027	0.002	0.018	0.008	0.027	<0.001
RatioVTI	0.218	0.086	0.351	0.002	0.233	0.111	0.356	<0.001

## Data Availability

The data presented in this study are available on request from the corresponding author. The data are not publicly available due to privacy reasons.

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
