# Peer review of "Association between Preoperative Retrograde Hepatic Vein Flow and Acute Kidney Injury after Cardiac Surgery [Author-notes fn1-diagnostics-12-00699]"

_diagnostics, 2022, doi:10.3390/diagnostics12030699_

Round 1

Reviewer 1 Report

The title of the article is “Association between preoperative retrograde hepatic vein flow and acute kidney injury after cardiac surgery”. The authors conducted a prospective, observational study. This study aimed to evaluate the association between preoperative hepatic venous flow patterns and the risk of AKI in patients after cardiac surgery.  This is an interesting paper. However, some of main important issues need to be verified to improve your work as following.

  1. The authors conducted a prospective, observational study. How the authors define censor strategies for prospective cohort including right, left, interval censor?
  2. For the multiple regression analysis, please clarify the methods for variables selection in model. For the linear regression analysis, the prediction is their objective the model assumptions as well as model performance, a test for the interaction between variables, multicollinearity, a test for the interaction between variables, and goodness-of-fit analysis should be performed and show in the results or supplementary. On the other hand, if the association or casual inference is their aim the confounding factors according to previous knowledge should be included to the model for appropriate effect estimation.

For variable and model selection, please refer to these articles:

1. Heinze G, Wallisch C, Dunkler D. Variable selection - A review and recommendations for the practicing statistician. Biometrical J [Internet]. 2018 May 1;60(3):431–49.

2. Vander Weele TJ. Principles of confounder selection. Eur J Epidemiol [Internet]. 2019 Mar 15;34(3):211–9.

3. Steyerberg EW, Vergouwe Y. Towards better clinical prediction models: seven steps for development and an ABCD for validation. Eur Heart J [Internet]. 2014 Aug 1;35(29):1925–31. Available from: www.r-project.org

  1. Include full details of how the authors handled missing data, outliers and include these in the results section. The author should elaborate about how you were dealing with that.
  2. Have you tested the associations of the non-linear variables in the regression model, e.g., with splines or a polynomial?
  3. How was sample size determined? The main concern is the small sample sizes of participants, which may have rendered it insufficiently powered to compare outcomes. There is no discussion on power calculation in the statistical methods section, so the reader is left unsure whether the number of participants was enough to achieve power to detect differences at z% statistical power. This may pose the risk of committing type I and type II errors. The results with too low statistical power will lead to weak conclusions about the meaning of the results. Statistical significance in a study does not mean that the main study is not required. In addition, the author should tone down conclusions to reflect the results.
  4. Finally, since I am not a native English user, I did not check for grammatical errors thoroughly. This should be done by an appropriate language reviewer.

Author Response

The title of the article is “Association between preoperative retrograde hepatic vein flow and acute kidney injury after cardiac surgery”. The authors conducted a prospective, observational study. This study aimed to evaluate the association between preoperative hepatic venous flow patterns and the risk of AKI in patients after cardiac surgery.  This is an interesting paper. However, some of main important issues need to be verified to improve your work as following

First, we would like to express our gratitude for the valuable comments of the reviewer. We have reanalyzed the available data and we have made and marked the changes in the manuscript.

  1. The authors conducted a prospective, observational study. How the authors define censor strategies for prospective cohort including right, left, interval censor?

We have used interval censoring in this study.

  1. For the multiple regression analysis, please clarify the methods for variables selection in model. For the linear regression analysis, the prediction is their objective the model assumptions as well as model performance, a test for the interaction between variables, multicollinearity, a test for the interaction between variables, and goodness-of-fit analysis should be performed and show in the results or supplementary. On the other hand, if the association or casual inference is their aim the confounding factors according to previous knowledge should be included to the model for appropriate effect estimation.

For variable and model selection, please refer to these articles:

  1. Heinze G, Wallisch C, Dunkler D. Variable selection - A review and recommendations for the practicing statistician. Biometrical J [Internet]. 2018 May 1;60(3):431–49.
  2. Vander Weele TJ. Principles of confounder selection. Eur J Epidemiol [Internet]. 2019 Mar 15;34(3):211–9.
  3. Steyerberg EW, Vergouwe Y. Towards better clinical prediction models: seven steps for development and an ABCD for validation. Eur Heart J [Internet]. 2014 Aug 1;35(29):1925–31. Available from: www.r-project.org

Response: We reanalyzed our data based on the reviewer’s suggestions. The following variables were considered including age, renal function, EuroSCORE II and AKI prediction score by Thakar et al. (J Am Soc Nephrol 2005;16:162–8.), operation time, fluid balance, and maximum vasopressor-inotropics score in the first 72 postoperative hours. The variables were selected based on a set of previously published studies with a focus on AKI after cardiac surgery. The multivariable model was tested for multicollinearity (by volume inflation factors) between these clinical variables. The adjusted changes in R2 were reported after each step of the regression model to determine the contribution of each additional variable that was added to the regression model.

Comment: Include full details of how the authors handled missing data, outliers and include these in the results section. The author should elaborate about how you were dealing with that.

Response: In this prospective study we had a very small number of missing information. The number (frequency) of missing data are shown in Table 1.

Comment: Have you tested the associations of the non-linear variables in the regression model, e.g., with splines or a polynomial?

Response: Continuous variables were first expanded with restricted cubic splines (to allow for potentially nonlinear effects), and were only used in linear form if the deviation from linearity was not significant as indicated by the global F test (P > 0.05). Our analyses indicated that AVmax, A VTI, and antero/reroVTI were non-linear variables, and therefore, these variables were transformed before entering them to the multivariable regressioin analyses.

Comment: How was sample size determined? The main concern is the small sample sizes of participants, which may have rendered it insufficiently powered to compare outcomes. There is no discussion on power calculation in the statistical methods section, so the reader is left unsure whether the number of participants was enough to achieve power to detect differences at z% statistical power. This may pose the risk of committing type I and type II errors. The results with too low statistical power will lead to weak conclusions about the meaning of the results. Statistical significance in a study does not mean that the main study is not required. In addition, the author should tone down conclusions to reflect the results.

Response: A power calculation section was added to the Methods section of the manuscript. In brief, our power analysis indicated that with 8 potential predictors of acute kidney injury a minimum of 90 patients were needed to perform a prediction model with adequate sample size and avoiding overfitting

Comment: Finally, since I am not a native English user, I did not check for grammatical errors thoroughly. This should be done by an appropriate language reviewer.

Response: An official English language reviewer through an official correction service was used to make sure that our manuscript is checked for grammatical errors.

Reviewer 2 Report

  1. Figure 1 is too small and difficult to read
  2. Did you do sample size calculation to determine the number of patients needed for enrollment?
  3. Please provide informed consent statement
  4. I am not why you collect MELD score when you excluded cirrhosis patients
  5. What is the definition of baseline creatinine?
  6. The definition of primary outcome – postoperative AKI is confusion. Normally, AKI is binary outcome (AKI vs. no AKI). I would suggest not to use “postoperative AKI” and just define primary outcomes as the percentage change of creatinine compared to baseline.
  7. In fact, I would suggest ot use KDIGO-based AKI as primary outcome as it is more clinically relevant. However, the adjusting analysis would be limited by the small number of outcomes (n=17).
  8. The association between hepatic wave and percentage change of creatinine was adjusted for patient- and surgery-related characteristic. Please elaborate adjusting covariates and explain the covariate selection
  9. As you used linear regression for analysis, did you test if the association between hepatic wave and percentage change in creatinine was in linear form
  10. “Univariate” should be “univariable”
  11. The result in table 3 would be overfitting because the limited number of AKI

Author Response

Review 2:

First of all we would like to thank you for your precise review. Please find our responses after your questions.

Comment: Figure 1 is too small and difficult to read.

Response: We have enlarged Figure 1.

Comment: Did you do sample size calculation to determine the number of patients needed for enrollment?

Response: We have done sample size calculation and we have added this para to the methods section. Our power analysis indicated that with 8 potential predictors of acute kidney injury a minimum of 90 patients were needed to perform a prediction model with adequate sample size and avoiding overfitting

Comment: Please provide informed consent statement.

Response: A statement was added to the Methods section.

Comment: I am not sure why you collect MELD score when you excluded cirrhosis patients.

Response: Our previous results suggested that MELD scores are useful quantitative markers of the preexisting liver function (Holndonner-Kirst E, Nagy A, Czobor NR, et al. Higher Transaminase Levels in the Postoperative Period After Orthotopic Heart Transplantation Are Associated With Worse Survival. J Cardiothorac Vasc Anesth. 2018;32(4):1711-1718. doi:10.1053/j.jvca.2018.01.002 ). Therefore, we assumed that the severity of the venous congestion, which we expressed by the retrograde/anterograde VTI ratios might be linked to discrete signs of liver congestion measured by MELD score.

Comment: What is the definition of baseline creatinine?

Response: Baseline creatinine was defined as the preoperative serum creatinine concentration measured after the indexed hospital admission but before surgery.

Comment: The definition of primary outcome – postoperative AKI is confusing. Normally, AKI is binary outcome (AKI vs. no AKI). I would suggest not to use “postoperative AKI” and just define primary outcomes as the percentage change of creatinine compared to baseline.

Response: A continuous AKI trait, the percentage change of the relative creatinine rise was used in our study to enhance the ability and power to identify risk factors. We have found that the KDIGO defined AKI criteria is not so informative in the identification of the incremental changes of the venous congestion. Two previous studies used the creatinine rise as primary outcome variable. In their analyses. they have found that even small relative rises in serum creatinine are associated with substantial reductions in postoperative event-free survival. (Stanford Smith Kidney International advance online publication, 17 June 2015; doi:10.1038/ki.2015.161; and Stafford-Smith M, Podgoreanu M, Swaminathan M et al. Association of genetic polymorphisms with risk of renal injury after coronary bypass graft surgery. Am J Kidney Dis 2005; 45: 519–530.)

Comment: In fact, I would suggest to use KDIGO-based AKI as primary outcome as it is more clinically relevant. However, the adjusting analysis would be limited by the small number of outcomes (n=17).

Response: Indeed, our first intention was the use KDIGO-based AKI as the primary outcome of the study, but the relatively small number of cases has prompted us to focus on a potential more sensitive measure of postoperative AKI.

Comment: The association between hepatic wave and percentage change of creatinine was adjusted for patient- and surgery-related characteristic. Please elaborate adjusting covariates and explain the covariate selection.

Response: We reanalyzed our data based on the reviewer’s suggestions. The following variables were considered including age, renal function, EuroSCORE II and AKI prediction score by Thakar et al. (J Am Soc Nephrol 2005;16:162–8.), operation time, fluid balance, amount of blood products and maximum vasopressor-inotropics score in the first 72 postoperative hours.These variables selection was based on a set of previously published studies with a focus on AKI after cardiac surgery.The multivariable model was tested for multicollinearity (by volume inflation factors) between these clinical variables. The adjusted changes in R2were reported after each step of the regression model to determine the contribution of each additional variable that was added to the regression model.

Comment: As you used linear regression for analysis, did you test if the association between hepatic wave and percentage change in creatinine was in linear form.

Response: Continuous variables were first expanded with restricted cubic splines (to allow for potentially nonlinear effects), and were only used in linear form if the deviation from linearity was not significant as indicated by the global F test (P > 0.05). Our analyses indicated that AVmax, A VTI, and antero/reroVTI were non-linear variables, and therefore, these variables were transformed before enetering them to the multivariable regressioin analyses.

Comment: “Univariate” should be “univariable”.

Response: We have revised our manuscript and “univariate” was changed to “univariable”

Comment: The result in table 3 would be overfitting because the limited number of AKI.

Response: We have removed this table from the revised version of the manuscript.

Reviewer 3 Report

This paper concerns an important issue: AKI incidence after cardiac surgery, and the authors put much effort into identifying a relatively noninvasive tool to help predict AKI risk. This is an interesting and useful paper. The authors carefully collected research data prospectively, and applied appropriate analytical techniques to discover messages in the data.

Although the VTI ratio of anterograde to retrograde waves only had a 0.654 discriminative value for the prediction of AKI, the multivariable logistic regression with the bootstrapping method confirmed that the ratio of retro/anterograde VTI was associated with the development of AKI after adjustment for euroSCORE and preoperative GFR values. Messages unearthed by the study do have value in practice, in addition to academic value.

The manuscript is well written. Nevertheless, the authors have to further check typos. For instance, Table 1 has an apparent typo regarding hNYHA III/IV.

Author Response

We would like to thank you for your kind comments. We have checked and corrected typos. 

Round 2

Reviewer 1 Report

The authors addressed all my previous concerns and significantly improved quality of the manuscript. I have no additional comment.

Author Response

Thank you for your comments!

Reviewer 2 Report

All of my comments have been properly addressed. 

Author Response

Thank you for your comments!